# Twice-Split Phosphorus Application Alleviates Low Temperature Stress by Improving Root Physiology and Phosphorus Accumulation, Translocation, and Partitioning in Wheat

**Hui Xu [1,2], Muhammad Ahmad Hassan [3] and Jincai Li [2,4,*]**

1  College of Landscape and Horticulture, Wuhu Institute of Technology, Wuhu 241003, China; aauxuhui@126.com
2  College of Agronomy, Anhui Agricultural University, Hefei 230036, China
3  Rice Research Institute, Anhui Academy of Agricultural Sciences, Hefei 230041, China; ahmaduaf93@stu.ahau.edu.cn
4  Jiangsu Collaborative Innovation Centre for Modern Crop Production, Nanjing 210095, China
*  Correspondence: lijc@ahau.edu.cn

**Abstract:** In the context of global warming, low temperature (LT) disasters in major crops are also becoming more common. LT stress in the Huang-Huai-Hai Plain, the central wheat region in China, caused a massive reduction in wheat yields. A step towards ensuring wheat yield stability and food security, this study investigated the effects of optimizing phosphorus application on the root physiology, dry matter phosphorus accumulation, translocation, and partitioning of wheat under LT stress, using the representative cultivar Yannong 19 as the test material. The treatments included conventional phosphorus application (R1) and twice-split phosphorus application (R2), followed by $-4\,^\circ C$ LT treatment and normal temperature (NT) treatment during the anther interval stage. Analysis of the root physiology (enzymatic activities and acid phosphatase, contents of malondialdehyde, soluble sugar, and soluble protein), phosphorus and dry matter accumulation, translocation, partitioning, and agronomic and yield-related components was carried during this research study. The results showed that the wheat root activity was significantly reduced and the antioxidant enzyme activities were increased to mitigate the damage of LT stress. Moreover, LT treatments damaged root function. The root activity and antioxidant properties were significantly lower than those of the NT treatment at the flowering stage. The dry matter and phosphorus accumulations were reduced by 30.6~33.6% and 15.1~21.3% at the flowering and maturity stages, resulting in final yield losses of 10.3~13.0%. In contrast, root activity increased by 16.1~27.2% in the twice-split phosphorus application treatments, and the root antioxidant characteristics were higher. As a result, dry matter and phosphorus accumulation increased after twice-split phosphorus application and their translocation to the grains was more; the final yield increased by 5.5~7.3%. Overall, the twice-split phosphorus application enhanced the physiological function of the root system and promoted the accumulation of nutrients and their transport to the grain, and alleviated the yield loss of wheat caused by LT stress.

**Keywords:** wheat; low temperature; optimizing phosphorus application; root physiology; phosphorus accumulation

## 1. Introduction

Population growth, climate change, and food security are critical issues humankind faces in the twenty-first century [1,2]. Wheat is one of the world's most widely grown crops, and its increased planted area and yield are vital for ensuring global food security and useful for reducing hunger [3]. In recent years, numerous natural disasters and abiotic stresses have accompanied environmental degradation and resource depletion [4].

Although the global average temperature is increasing yearly and many studies have been conducted on the effects of high temperatures on crop production, the frequency and intensity of low temperature (LT) stress are undoubtedly increasing [5,6]. As a result, researchers in many nations have started to shed light on the influence of LT stress on the yield of crops, mainly the frequency and intensity of LT stress, which has had a significant impact on wheat production in Europe, North America, and China [7–9].

The Huang-Huai-Hai Plain, China's greatest wheat-producing region, is situated in a monsoon climatic zone and it is vulnerable to the effect of cold air from the northwest in the spring [10]. The anther interval stage of wheat is between the jointing and booting stages, which is also the critical stage of developing young spikes [11]. LT stress can cause imbalances in wheat plants' source−sink relationships, including reduced root absorptive function, leaf photosynthetic capacity, and limited development of young spikes [4]. LT stress also inhibits root respiration and metabolic activities in wheat, affecting its ability to absorb nutrients and water [12]. Numerous studies have demonstrated that the formation of reactive oxygen species (ROS) and disruption of cell membranes are the primary sources of damage to wheat plants [13–15]. Wheat plants under LT stress had significantly higher levels of ROS and membrane lipid peroxide, i.e., malondialdehyde (MDA) content, as well as higher antioxidant enzyme activity levels, which improved the plants' ability to withstand the stress [16]. However, as the degree of LT deepens, it can cause ROS metabolic imbalance, damage the antioxidant system, and it ruptures the cell membrane structure, thereby seriously affecting wheat plants' normal growth and development [4]. The effects of LT stress on wheat plants are mainly focused on leaves and spikes, while very few studies have been conducted on the root system.

Phosphorus is one of the major nutrients required by plants, and it plays a vital role in plant growth, development, and optimal metabolic functioning [17,18]. Phosphorus, a nonrenewable resource, is predicted to exceed world phosphorus demand and supply by 2045 [19]. Phosphorus enhances crop adaptation to the external environment by participating in signal transduction, energy dynamics, and enzyme catalysis [20]. Phosphorus application increases water and nutrient uptake by the root system [21–23]. Adequate phosphorus supply at the seedling stage is conducive to developing strong seedlings, which not only improves the LT resistance of plants, but also promotes the transport of nutrients to the grains after the flowering stage [4,24]. Rafiullah et al. [25] pointed out that the stage of jointing to booting is the critical stage of phosphorus deficiency in wheat, and optimizing phosphorus application can increase soil phosphorus levels and promote the absorption and utilization of phosphorus in wheat plants. When wheat plants are under abiotic stress such as LT stress, optimizing phosphorus application can reduce yield loss by improving leaf photosynthesis and dry matter accumulation [26,27]. Under LT stress, phosphorus can increase the activity of enzymes related to sucrose metabolism and promote sucrose accumulation in young spikes, thus improving spikelet development and grain setting [10]. Based on the results of previous studies, optimizing phosphorus application can be an effective means to cope with LT stress in wheat [4,10,26]. However, it is uncertain whether optimizing phosphorus application (i.e., twice-split phosphorus application treatment) during the anther interval stage can mitigate the detrimental impacts of LT stress on root physiology and phosphorus accumulation, translocation, and partitioning in wheat.

In the present study, we hypothesized that twice-split phosphorus application would enhance wheat resilience to LT stress during the anther interval stage, improve root physiology, and increase phosphorus accumulation and final yield. The objectives of this study were to (1) examine the effects of twice-split phosphorus application on wheat roots' physiological indicators, such as antioxidant capacities under LT stress during the anther interval stage; (2) focus on the effects of various treatments on the accumulation, translocation, and distribution of dry matter and phosphorus in wheat plants; and (3) explain the beneficial effects of twice-split phosphorus application on wheat yield and its components. We believe that this study will provide a theoretical basis and technical support for the extension and

application of twice-split phosphorus application to cope with LT stress in field wheat production.

## 2. Materials and Methods

### 2.1. Experimental Site

A pot experiment was conducted in 2021–2022 during the growing season of wheat at the Nongcui Garden of Anhui Agricultural University, Hefei, Anhui Province, China (31°86′ N, 117°26′ E). Experimental soil was taken from the 0~20 cm tillage layer with an organic matter content of 15.6 g kg$^{-1}$ and available phosphorus, nitrogen, and potassium contents of 25.8, 133.7, and 204.3 mg kg$^{-1}$ this year, respectively. Each pot (a diameter of 26 cm and a height of 35 cm) was filled with 10 kg of soil on 11 November 2021, while the wheat seeds were sown.

### 2.2. Experimental Design

The wheat cultivar Yannong 19 (YN19), widely grown in the Huang-Huai-Hai wheat areas, was used as the test cultivar. The experiment was a randomized block design, and the phosphorus application methods were conventional phosphorus application (all basal application (R1)) and twice-split phosphorus application (50% each at the pre-sowing and jointing stage (R2)). Two temperature levels were set: normal temperature treatment (NT, 10 °C) and LT treatment (−4 °C). Wheat pots were fertilized according to twice the amount of fertilizer applied to the field, with 1.8 g of urea applied to each pot during the whole life cycle, of which 1.2 g was applied before sowing, and 1.7 g of potassium sulphate was applied all basally. The conventional phosphorus application treatment was 5.0 g of superphosphate (active ingredient percentage $\geq$ 12.0%) applied basally, and the twice-split phosphorus application treatment was 2.5 g of superphosphate applied basally before sowing and 2.5 g of superphosphate applied basally at the jointing stage. In this experiment, 72 wheat pots were planted, meaning 18 pots per treatment.

Half of the potted plants were transferred to an artificial climate chamber (DGXM-1008, Ningbo Jiangnan Instrument Manufacturing Factory, Ningbo, China) during the anther interval stage. The LT treatment was set at −4 °C in the artificial climate chamber with a humidity of 75% and light intensity of 0 μmol·m$^{-2}$·s$^{-1}$·s from 01:00 a.m. to 05:00 a.m. The other half of the potted plants were placed in the field at normal temperature, averaging 10 °C from 01:00 a.m. to 05:00 a.m. in 2022. After LT treatment, the wheat pots were moved back into the field and grown to maturity. Other management practices were the same as those used in general high-yielding areas.

### 2.3. Sampling and Measurement

2.3.1. Root Physiology

The wheat root's physiological properties were evaluated after LT treatment and at the flowering stage. The wheat root samples were frozen in liquid nitrogen and then sealed in an ultra-low temperature refrigerator with tin foil to protect them from light. The MDA content and the enzymatic activities of SOD, CAT, and POD of three wheat root samples were measured according to the assay of Li et al. [28]. The wheat root acid phosphatase activity, root activity, soluble sugar, and soluble protein content were measured using the method of Xu et al. [10].

2.3.2. Dry Matter Accumulation, Translocation, and Partitioning in Wheat Plants

The aboveground portion of three random wheat plants was sampled at the flowering and maturity stages and placed in an oven for weighing the dry matter. The plant was divided into three organs at the flowering stage, spike, stalk + leaf sheath, and leaf, and four organs at maturity, grains, rachis + glumes, stalk + leaf sheath, and leaf. Here, leaf and stalk + leaf sheath were considered as vegetative organs. The pre-flowering vegetative organ dry matter translocation (DMT), pre-flowering vegetative organ dry matter translocation rate (PDMT), contribution of dry matter transportation of vegetative organs before the

flowering stage to dry matter of grains (CDMT), dry matter accumulation of vegetative organs after flowering stage (DMAF), and contribution of dry matter after flowering stage to dry matter of grains (CDMAF) were calculated according to Xu et al. [26] and Shi et al. [29].

### 2.3.3. Phosphorus Accumulation, Translocation, and Partitioning in Wheat Plants

Three random wheat samples at the flowering and maturity stages were dried and pulverized by a ball mill (MM400, Retsch Company, Arzberg, Germany) and then digested using the $H_2SO_4$-$H_2O_2$ method [30]. A fully automated continuous flow analyzer determined the phosphorus content in the digest (AA3, Seal Company, Norderstedt, Germany). Moreover, the following parameters related to phosphorus accumulation and remobilization within the wheat plants during grain filling were calculated [31]:

(1)    Wheat plant phosphorus accumulation = Plant dry matter weight $\times$ phosphorus content.
(2)    Phosphorus translocation before flowering stage (PT) = Vegetative organ phosphorus accumulation at flowering stage $-$ Vegetative organ phosphorus accumulation at maturity stage.
(3)    Phosphorus translocation rate before flowering stage (PTR) = PT $\div$ Vegetative organ phosphorus accumulation at flowering stage $\times$ 100%.
(4)    Phosphorus translocation contribution rate before flowering stage to grains (PTCG) = PT $\div$ Grain phosphorus accumulation at maturity $\times$ 100%.
(5)    Phosphorus accumulation after flowering stage (PAAF) = Phosphorus accumulation at maturity $-$ Phosphorus accumulation at flowering stage.
(6)    Phosphorus accumulation contribution rate after flowering stage to grains (PACG) = 100% $-$ PTCG.
(7)    Phosphorus harvest index (PHI) = Grains phosphorus accumulation at maturity $\div$ wheat plant phosphorus accumulation at maturity $\times$ 100%.

### 2.3.4. Yield and Its Components

The spike number per plant of wheat was investigated one week before the maturity stage, and three representative wheat plants were selected from each treatment at maturity to measure the grain number per spike and 1000-grain weight, thus calculating the final yield.

### *2.4. Statistical Analysis*

Data were analyzed by ANOVA using SPSS 19.0 software. The least significant difference method was used to test the significance of the differences ($p = 0.05$) between treatments and the results were expressed as mean $\pm$ standard error.

## 3. Results

### *3.1. Antioxidant Enzyme Activities and MDA Content in Root*

As shown in Figure 1, the SOD and CAT activities in wheat roots after LT stress were significantly increased on the day of low temperature treatment (LTT) and the POD activity was not significantly increased. At the flowering stage, antioxidant enzyme changes showed the opposite trends to those at LTT, with all of the treatments of LT stress being lower than those of NT. Moreover, the twice-split phosphorus application increased the antioxidant enzyme activities at the LTT and flowering stages. At LTT, the SOD activities of the LT treatments (R1LT and R2LT) were increased by 18.7% and 31.2%, respectively, compared with R1NT, whereas the twice-split phosphorus application treatments (R2NT and R2LT) were increased by 13.6% and 10.5%, respectively, compared with the conventional phosphorus application treatments (R1NT and R1LT) at the same temperature. Similarly, the POD and CAT activities of the LT treatments were increased by 16.7~38.4% and 37.2~64.6%, respectively, compared with R1NT, whereas the twice-split phosphorus application treatments were increased by 13.0~18.6% and 20.0~24.7%, respectively, compared with the conventional phosphorus application treatments. At the flowering stage, the SOD, POD,

and CAT activities of the LT treatments were decreased by 4.4~14.6%, 4.4~23.9%, and 3.0~21.7%, respectively, compared with R1NT, whereas the twice-split phosphorus application treatments were increased by 7.4~12.0%, 17.6~25.6%, and 23.8~31.5%, respectively, compared with the conventional phosphorus application treatments.

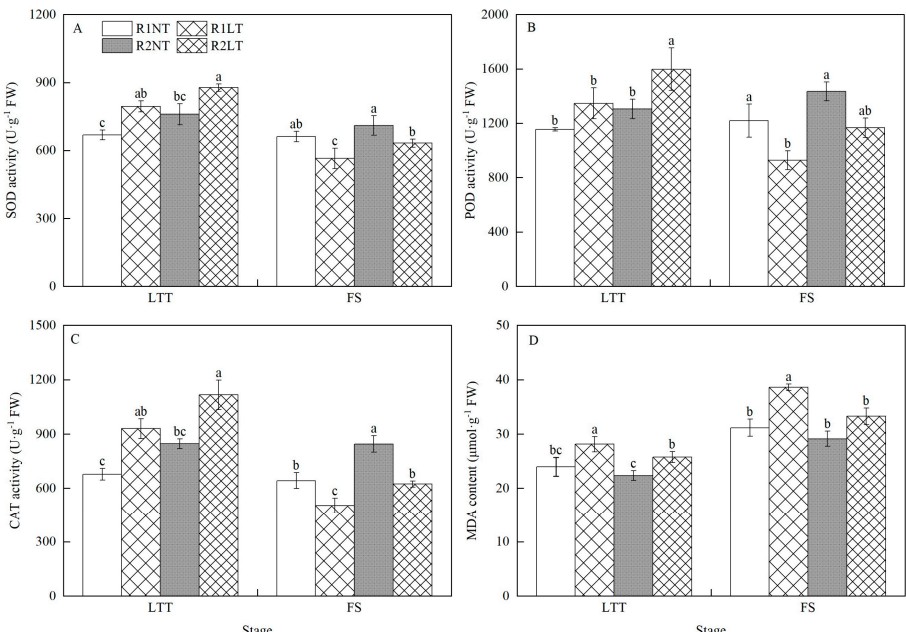

**Figure 1.** Effects of twice-split phosphorus application on the activities of superoxide dismutase (SOD) (**A**), peroxidase (POD) (**B**), catalase (CAT) (**C**), and malondialdehyde (MDA) (**D**) content after LT treatment. R1, R2, NT, LT, LTT, and FS represent conventional phosphorus application, twice-split phosphorus application, normal temperature, low temperature, day of low temperature treatment, and flowering stage, respectively. Different lowercase letters indicate significant differences between treatments ($p < 0.05$). Vertical bars represent the standard error of the mean.

The MDA content in the wheat roots was significantly higher under LT treatments and decreased after a twice-split phosphorus application. At the LTT and flowering stage, the MDA content of the LT treatments was increased by 7.3~17.3% and 6.6~23.9%, respectively, compared with R1NT, whereas the twice-split phosphorus application treatments were decreased by 7.0~8.5% and 6.5~13.9%, respectively, compared with the conventional phosphorus application treatments at the same temperature.

The twice-split phosphorus application increased the antioxidant enzyme activities and decreased the accumulation of MDA in the wheat root.

### 3.2. Acid Phosphatase Activity, Root Activity, Soluble Sugar, and Soluble Protein Content in Root

ACP in wheat roots was significantly reduced by 27.1~36.0% after LT stress at LTT increasing by 3.4~15.2% after LT stress compared with R1NT at the flowering stage. The twice-split phosphorus application treatments were reduced compared with conventional phosphorus application treatments by 10.8~12.2% and 10.2~10.5% (Figure 2A).

The root activity showed the same trend in the LLT and flowering stage. The root activity decreased by 7.0~26.9% and 24.6~39.1% after LT stress, whereas the twice-split phosphorus application treatments maintained a higher (16.1~27.2%) root activity (Figure 2B).

Figure 2C,D show that the SS and SP contents followed the same change trend as the antioxidant enzymes in the LLT and flowering stage. At LTT, the SS and SP contents of the LT treatments increased by 34.9~70.5% and 15.4~33.2%, respectively, compared with R1NT, whereas the twice-split phosphorus application treatments increased by 22.8~26.3% and 6.8~15.4%, respectively, compared with the conventional phosphorus application treatments. In the flowering stage, the SS and SP contents of the LT treatments were

decreased by 2.5~12.4% and 0~17.8%, respectively, whereas the twice-split phosphorus application treatments were increased by 16.5~17.5% and 17.7~22.8%, respectively.

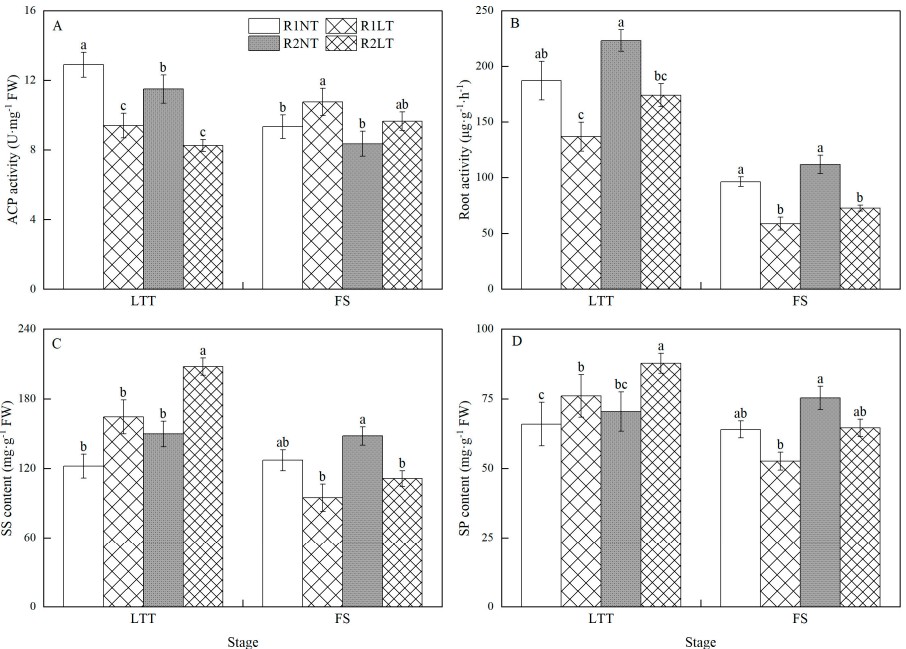

**Figure 2.** Effects of twice-split phosphorus application on acid phosphatase activity (ACP) (**A**), root activity (**B**), soluble sugar (SS) (**C**), and soluble protein (SP) (**D**) content after LT treatment. R1, R2, NT, LT, LTT, and FS represent conventional phosphorus application, twice-split phosphorus application, normal temperature, low temperature the day of low temperature treatment, and flowering stage, respectively. Different lowercase letters indicate significant differences between treatments ($p < 0.05$). Vertical bars represent the standard error of the mean.

### 3.3. Dry Matter Accumulation and Partitioning at the Flowering and Maturity Stages of Wheat Plants

As shown in Figure 3, the dry matter weight increased from the flowering stage to the maturity stage, while the dry matter weight of wheat plants and different organs decreased significantly after LT stress. The dry matter weight of wheat plants was reduced considerably by 27.1~29.8% and 30.6~33.6% after LT stress at the flowering and maturity stages, respectively. The dry matter weight in the twice-split phosphorus application treatments was increased compared with conventional phosphorus application treatments by 1.2~3.8% and 3.2~4.5%, respectively. In addition, the dry matter weight of different organs was higher in the twice-split phosphorus application treatments than in the conventional phosphorus application treatments at the same temperature.

At the flowering stage, dry matter was mainly accumulated in the vegetative organs, accounting for 80.2~80.9% (Figure 4A). At the maturity stage, the dry matter weight in the vegetative organs was substantially reduced compared with the flowering stage, whereas the dry matter in the grains was higher than that accumulated in the vegetative organs in both the R1NT and R2NT treatments (Figure 4B).

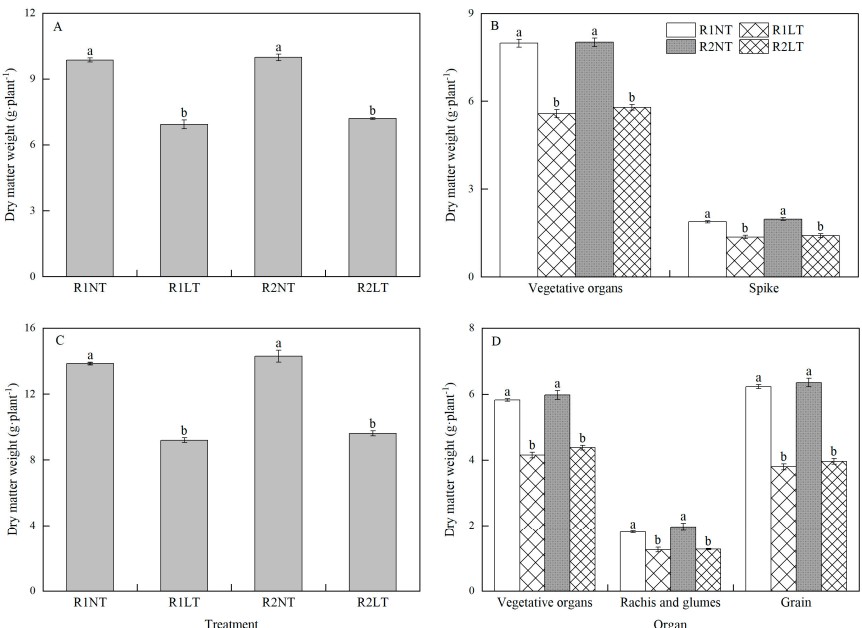

**Figure 3.** Effects of twice-split phosphorus application on dry matter weight in the flowering and maturity stages after LT treatment. (**A**) Dry matter weight in the flowering stage. (**B**) Dry matter weight of different organs in the flowering stage. (**C**) Dry matter weight in the maturity stage. (**D**) Dry matter weight of different organs in the maturity stage. R1, R2, NT, and LT represent conventional phosphorus application, twice-split phosphorus application, normal temperature, and low temperature, respectively. Different lowercase letters indicate significant differences between treatments ($p < 0.05$). Vertical bars represent the standard error of the mean.

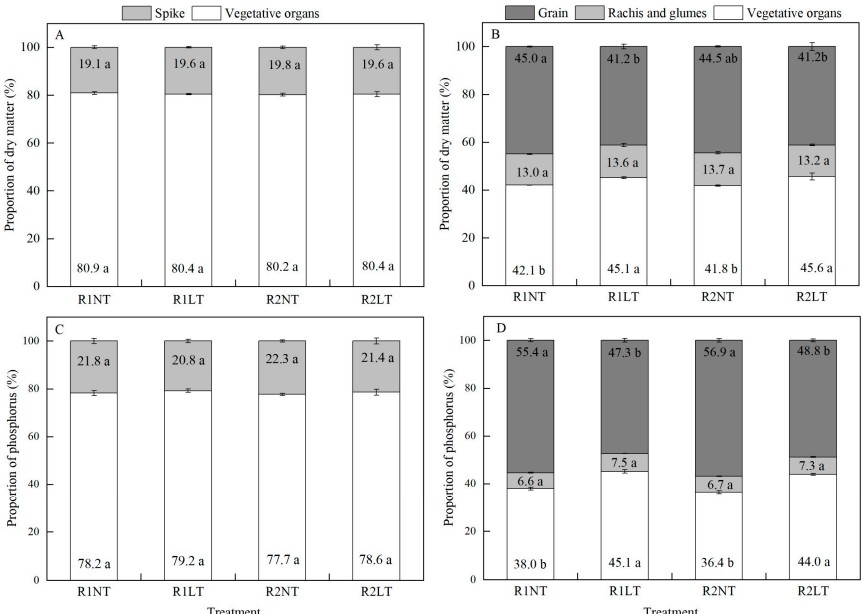

**Figure 4.** Effects of twice-split phosphorus application on dry matter and phosphorus partitioning of different organs in the flowering and maturity stages after LT treatment. (**A**) Proportion of dry matter weight in the flowering stage. (**B**) Proportion of dry matter weight in the maturity stage. (**C**) Proportion of phosphorus accumulation in the flowering stage. (**D**) Proportion of phosphorus accumulation in the maturity stage. R1, R2, NT, and LT represent conventional phosphorus application, twice-split phosphorus application, normal temperature, and low temperature, respectively. Different lowercase letters indicate significant differences between treatments ($p < 0.05$). Vertical bars represent the standard error of the mean.

### 3.4. Phosphorus Accumulation and Partitioning at Flowering and Maturity Stages of Wheat Plants

As shown in Figure 5, phosphorus accumulation increased from the flowering stage to the maturity stage. In contrast, phosphorus accumulation of wheat plants and different organs in the flowering stage decreased significantly after LT stress. Phosphorus accumulation of wheat plants was reduced considerably by 14.1~18.6% and 15.1~21.3% after LT stress in the flowering and maturity stages, respectively. Phosphorus accumulation in the twice-split phosphorus application treatments was increased compared with conventional phosphorus application treatments by 3.6~5.5% and 5.7~7.9%, respectively. Similarly, phosphorus accumulation of different organs was higher in the twice-split phosphorus application treatments than in conventional phosphorus application treatments at the same temperature.

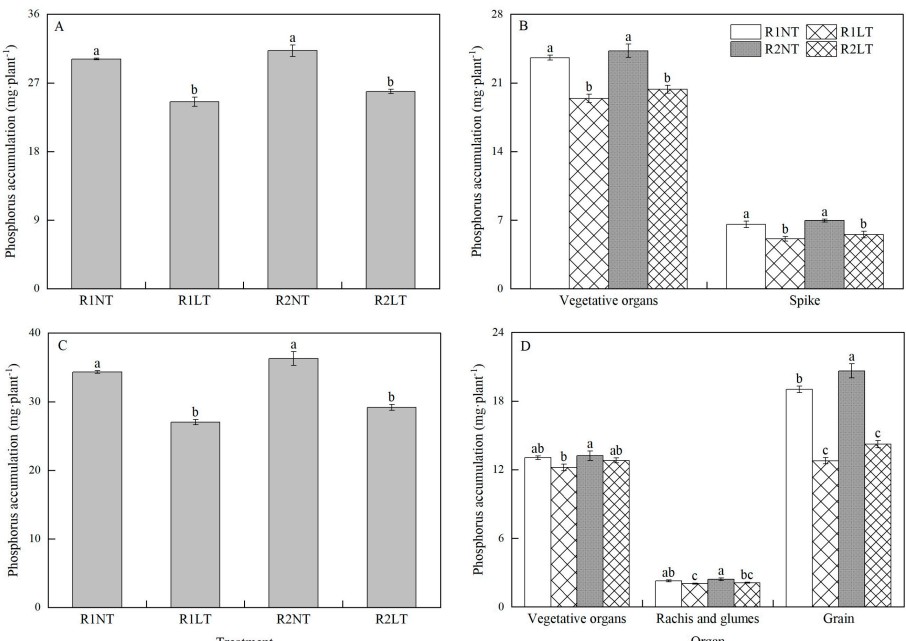

**Figure 5.** Effects of twice-split phosphorus application on phosphorus accumulation in the flowering and maturity stages after LT treatment. (**A**) Phosphorus accumulation in the flowering stage. (**B**) Phosphorus accumulation of different organs in the flowering stage. (**C**) Phosphorus accumulation in the maturity stage. (**D**) Phosphorus accumulation of different organs in the maturity stage. R1, R2, NT, and LT represent conventional phosphorus application, twice-split phosphorus application, normal temperature, and low temperature, respectively. Different lowercase letters indicate significant differences between treatments ($p < 0.05$). Vertical bars represent the standard error of the mean.

In the flowering stage, phosphorus was mainly accumulated in the vegetative organs, accounting for 77.7~79.2% (Figure 4C). In the maturity stage, the phosphorus accumulation in the vegetative organs was reduced by 34.1~41.3% compared with the flowering stage, and phosphorus accumulation in the grains was higher than in the vegetative organs (Figure 4D).

### 3.5. Dry Matter Translocation at Flowering and Maturity Stages of Wheat Plants

LT stress reduced DMT, PDMT, DMAF, and CDMAF, while CDMT increased. There was no significant difference between PDMT, CDMT, and CDMAF treatments (Table 1). The twice-split phosphorus application could reduce DMT, PDMT, and CDMT compared with the conventional phosphorus application, while DMAF and CDMAF increased after the twice-split phosphorus application. DMT and DMAF significantly decreased after LT stress by 34.2~34.9% and 37.2~41.7%, respectively. DMT decreased by 1.0~5.7% and DMAF increased by 6.1~7.7% after the twice-split phosphorus application. CDMT increased by

2.5~8.1% under LT stress, which decreased by 5.2~7.4% in the twice-split phosphorus application treatments than in the conventional phosphorus application treatments at the same temperature. On the contrary, CDMAF decreased by 1.3~4.3% under LT stress and increased by 3.1~3.9% in the twice-split phosphorus application treatments.

**Table 1.** Effects of twice-split phosphorus application on dry matter translocation in wheat plants after LT treatment.

| Treatment | Dry Matter before Flowering Stage | | | Dry Matter after Flowering Stage | |
|---|---|---|---|---|---|
| | DMT (g·Plant$^{-1}$) | PDMT (%) | CDMT (%) | DMAF (g·plant$^{-1}$) | CDMAF (%) |
| R1NT | 2.16 ± 0.11 a | 27.0 ± 0.9 a | 34.6 ± 1.5 a | 4.07 ± 0.07 a | 65.4 ± 1.5 a |
| R1LT | 1.42 ± 0.06 b | 25.5 ± 0.4 a | 37.4 ± 1.6 a | 2.37 ± 0.09 b | 62.6 ± 1.6 a |
| R2NT | 2.04 ± 0.09 a | 25.4 ± 1.0 a | 32.1 ± 1.6 a | 4.32 ± 0.15 a | 67.9 ± 1.6 a |
| R2LT | 1.41 ± 0.05 b | 24.3 ± 0.7 a | 35.5 ± 1.5 a | 2.56 ± 0.10 b | 64.5 ± 1.5 a |

R1, R2, NT, and LT represent conventional phosphorus application, twice-split phosphorus application, normal temperature, and low temperature, respectively. DMT, the pre-flowering vegetative organ dry matter translocation; PDMT, pre-flowering vegetative organ dry matter translocation rate; CDMT, the contribution of dry matter transportation of vegetative organs before the flowering stage to dry matter of grains; DMAF, dry matter accumulation of vegetative organs after flowering stage; CDMAF, contribution of dry matter after flowering stage to dry matter of grains. Different lowercase letters following the data in the same column indicate significant differences ($p < 0.05$). Values are means ± standard error ($n = 3$).

### 3.6. Phosphorus Translocation at Flowering and Maturity Stages of Wheat Plants

LT stress reduced PT, PTR, PAAF, and PHI, while the RILT treatment of PTCG and the R2LT treatment of PACG increased compared with R1NT (Table 2). Twice-split phosphorus application increased PT, PAAF, PACG, and PHI compared with conventional phosphorus treatment, while PTCG decreased slightly after twice-split phosphorus application. PT and PAAF significantly decreased by 28.5~34.1% and 21.8~40.7% after LT stress, respectively. PT and PAAF increased by 4.2~4.9% and 20.6~31.8% after the twice-split phosphorus application. PTCG decreased by 2.0~4.4%, and PACG increased by 4.2~8.1% after the twice-split phosphorus application. Moreover, PHI significantly decreased by 11.9~14.5% after LT stress and increased by 2.7~3.1% under the twice-split phosphorus application treatments.

**Table 2.** Effects of twice-split phosphorus application on phosphorus translocation in wheat plants after LT treatment.

| Treatment | Phosphorus before Flowering Stage | | | Phosphorus after Flowering Stage | | PHI (%) |
|---|---|---|---|---|---|---|
| | PT (mg·Plant$^{-1}$) | PTR (%) | PTCG (%) | PAAF (mg·Plant$^{-1}$) | PACG (%) | |
| R1NT | 10.53 ± 0.32 a | 44.7 ± 1.0 a | 55.4 ± 1.1 a | 4.20 ± 0.30 ab | 44.6 ± 1.1 a | 55.4 ± 0.7 a |
| R1LT | 7.23 ± 0.54 b | 37.2 ± 2.1 b | 56.5 ± 3.2 a | 2.49 ± 0.32 c | 43.5 ± 3.2 a | 47.3 ± 0.8 b |
| R2NT | 11.05 ± 0.51 a | 45.5 ± 1.3 a | 53.5 ± 1.2 a | 5.07 ± 0.41 a | 46.5 ± 1.2 a | 56.9 ± 0.8 a |
| R2LT | 7.53 ± 0.20 b | 37.0 ± 0.3 b | 52.9 ± 1.2 a | 3.28 ± 0.22 bc | 47.1 ± 1.2 a | 48.8 ± 0.5 b |

R1, R2, NT, and LT represent conventional phosphorus application, twice-split phosphorus application, normal temperature, and low temperature, respectively. PT, phosphorus translocation before the flowering stage; PTR, phosphorus translocation rate before the flowering stage; PTCG, phosphorus translocation contribution rate before flowering stage to grains; PAAF, phosphorus accumulation after flowering stage; PACG, phosphorus accumulation contribution rate after flowering stage to grains; PHI, phosphorus harvest index. Different lowercase letters following the data in the same column indicate significant differences ($p < 0.05$). Values are means ± standard error ($n = 3$).

### 3.7. Yield and Its Components in Wheat

Overall, LT stress significantly reduced the wheat yield by 31.9~36.6% in the R1LT and R2LT compared with the R1NT treatment, whereas twice-split phosphorus application treatments ensured higher (5.5~7.3%) yields compared with the conventional phosphorus application treatments (Figure 6). The trends of the spike number, grains number per spike,

and 1000-grain weight were generally consistent with those of the yield. The spike number, grains number per spike, and 1000-grain weight of the LT treatments were decreased by 12.0~14.8%, 14.0~14.6%, and 10.3~13.0%, respectively, compared with R1NT, whereas the twice-split phosphorus application treatments were increased by 3.3~3.7%, 0~0.8% and 1.7~3.1%, respectively, compared with the conventional phosphorus application treatments.

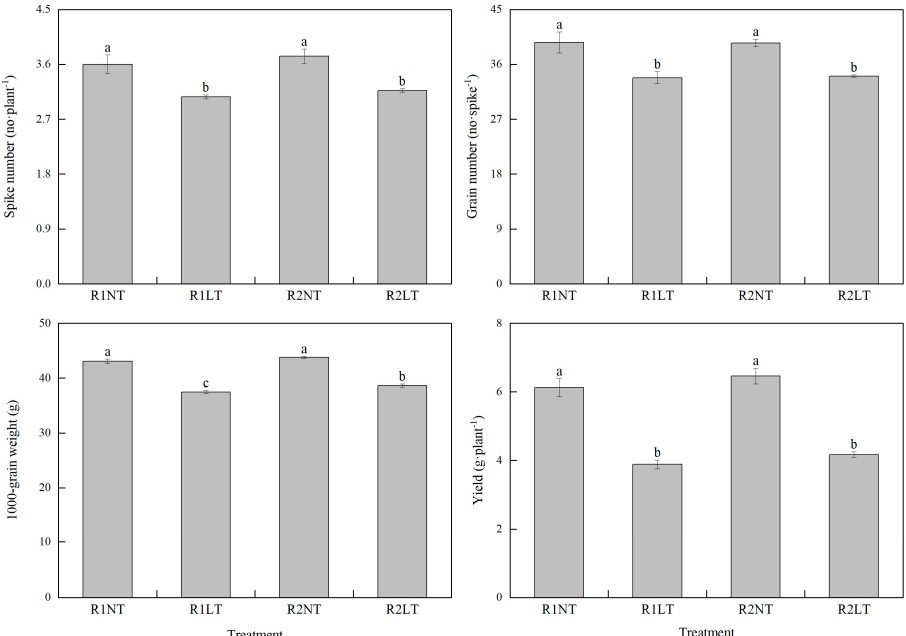

**Figure 6.** Effects of twice-split phosphorus application on yield and its components after LT treatment. R1, R2, NT, and LT represent conventional phosphorus application, twice-split phosphorus application, normal temperature, and low temperature, respectively. Different lowercase letters indicate significant differences between treatments (*p* < 0.05). Vertical bars represent the standard error of the mean.

## 4. Discussion

### 4.1. Effects of Optimizing Phosphorus Application on Root Physiology

The root is an indispensable organ for wheat growth and development, directly affecting affecting wheat aboveground yield formation and stress resistance [32]. Wheat root activity and ACP are important in mineral nutrient and water uptake and conductance. As a crucial hydrolytic enzyme, ACP is involved in phosphorus catabolism and reutilization [33]. In this study, we showed that ACP activity in wheat root was significantly reduced by LT stress, and ACP activity was further reduced under twice-split phosphorus application treatments. This was because the twice-split phosphorus application provided a sufficient phosphorus environment for the wheat root, and thus the ACP activity was suppressed [34]. The root activity significantly decreased after LT stress, and the root activity of each treatment increased by 16.1~27.2% after the twice-split phosphorus application. This may be due to the improvement in soil microbial activity through optimizing phosphorus applications [4]. Similarly, previous studies have indicated that delaying the application time of slow-release fertilizer increased the soil's enzyme activity, fertility, and root activity [35].

LT stress disrupts the antioxidant system of the crop, resulting in MDA accumulation and toxicity [36]. At the same time, crops have evolved a range of physiology and metabolism to mitigate this damage in their natural environment [37,38]. Along with LT stress and increased MDA content, crop plants scavenge reactive oxygen radicals through the antioxidant system to protect cells and improve plant LT tolerance [7,39]. We found that the antioxidant enzyme activities in the root system of all treatments of twice-split phosphorus applications were increased to different degrees. As a result, the MDA content

in the root system was reduced by 7.0–8.5% and 6.5–13.9% at the LTT and flowering stages, respectively. In addition, the accumulation of SS and SP also improved wheat plants' LT tolerance, which was consistent with previous studies [7,12]. Meanwhile, the antioxidant enzyme activity under LT stress increased at LTT and decreased in the flowering stage. This is mainly due to the increased activity of antioxidant enzymes in wheat plants after LTT stress to alleviate damage [4]. Due to abiotic stress accelerating wheat plants senescence, antioxidant enzyme activity decreases in the late growing periods [40].

### 4.2. Effects of Optimizing Phosphorus Application on Dry Matter and Phosphorus Accumulation, Translocation, and Partitioning

Accumulation, translocation, and partitioning of dry matter and nutrients are the basis for yield formation in wheat [26]. Under LT stress, dry matter and nutrient accumulation in wheat plants were significantly decreased due to the limited capacity of the root system for absorption and photosynthesis [4,5]. Among them, LT stress ($-6\sim-2$ °C) at the jointing and booting stages resulted in 17.8~35.9% and 20.9~43.7% reduction in the aboveground dry matter accumulation of wheat, respectively [5]. In our study, dry matter and phosphorus accumulation at maturity were reduced by 30.6~33.6% and 15.1~21.3% after LT stress, respectively, whereas twice-split phosphorus application treatments were relatively elevated by 3.2~4.5% and 5.7~7.9%, respectively. Liu et al. [41] indicated that optimizing phosphorus application improved soil fertility, shoot biomass, and phosphorus use efficiency and aboveground phosphorus accumulation increased by 13~14%. This may be because optimizing phosphorus application increased the phosphorus availability in the soil phosphorus pool, which could meet the nutritional demand for wheat growth and development [18]. It has been shown that the application of additional urea before LT stress at the seedling stage of wheat was conducive to promoting the uptake of fertilizers by the root system, increasing the aboveground dry matter weight, and restoring wheat growth [42].

Similarly, the present study showed that twice-split phosphorus application treatments increased dry matter and phosphorus accumulation, and the translocation and partitioning to grains also enhanced after flowering, with PHI increasing by 2.7~3.1% in the twice-split phosphorus application treatments than that in the conventional phosphorus application treatments at the same temperature. Wang et al. [43] pointed out that effective phosphorus homeostasis in plants can be achieved through phosphorus storage and redistributive utilization among different organs, and senescent leaves can serve as a major source of nutrients such as phosphorus during the later stages of wheat plant nutrient and reproductive growth [44–46].

### 4.3. Effects of Optimizing Phosphorus Application on Yield and Its Components

Phosphorus uptake and crop utilization are vital for determining the final crop yield [47–49], and optimizing phosphorus application can be a suitable agronomic and economical means of improving crop yield and nutrient accumulation, contributing to food and nutritional security [50,51]. LT stress causes source−sink relationship imbalance in the plant, limiting the growth and development of wheat and, hence, yield losses [4,52,53]. In previous studies, LT stress reduced wheat yield to varying degrees, with substantial LT stress even causing more than 50% yield reduction [54–56]. The main reasons for yield loss include decreased leaf photosynthetic capacity, reduced dry matter accumulation, and spikelet degradation [57–59]. Our study showed that 1000-grain weight and final yield were reduced by 31.9~36.6% and 10.3~13.0% under LT treatments due to reduced accumulation and translocation of dry matter and phosphorus to the grain. Many studies have also demonstrated that optimizing phosphorus application ensures adequate phosphorus supply, normal wheat growth, and stable wheat yield by increasing the number of fertile spikelets and 1000-grain weight [10,27]. Our study mainly revealed that the twice-split phosphorus application reduced yield loss due to LT stress by enhancing the root uptake capacity and nutrient partitioning. Twice-split phosphorus application also

increased the 1000-grain weight and final yield by 1.7~3.1% and 5.5~7.3%, respectively, relative to conventional phosphorus application treatments. Notably, the combination of phosphorus-solubilizing bacteria and foliar spraying can increase phosphorus uptake by wheat plants and promote wheat yield and quality [28,60,61]. This is because phosphorus fertilizer, an important nutrient for wheat growth and development, can also reduce spikelet degradation and increase spikelet setting by improving the antioxidant properties of young spikes [4].

## 5. Conclusions

This study concluded that LT stress reduces the nutritional source of wheat plants by disrupting root physiology. Therefore, dry matter and phosphorus accumulation significantly decrease during both the flowering and maturity stages, decreasing the final yield. The twice-split phosphorus application increased the absorption and utilization of phosphorus by wheat, improved the root activity and antioxidant capacity, and alleviated the premature senescence of wheat plants. In addition, the accumulation, translocation, and partitioning of dry matter and phosphorus to grains increased after the flowering stage, providing a basis for an increase in yield and composition.

This research will offer wheat farmers practical insights to mitigate abiotic stress and improve irrigation and fertilizer management at critical phases. In future research, it is necessary to explore measures to improve the combination of fertilizer effectiveness and wheat stress tolerance.

**Author Contributions:** Conceptualization, H.X.; formal analysis, H.X.; methodology, H.X.; resources, J.L.; software, H.X.; writing—original draft, H.X. and M.A.H.; writing—review and editing, H.X. All authors have read and agreed to the published version of the manuscript.

**Funding:** The work was supported by the Major Science and Technology Projects in Anhui Province, China (202003b06020021); the Natural Science Foundation of Anhui Province, China (2008085QC122); the Postgraduate Quality Engineering Project in Anhui Province, China (2022cxcysj066); and the Special Fund for Anhui Agriculture Research System, China.

**Data Availability Statement:** Not applicable.

**Acknowledgments:** We thank all members of the Crop Physiology and Ecology Laboratory (College of Agronomy, Anhui Agricultural University) for their assistance.

**Conflicts of Interest:** The authors declare no conflict of interest.

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
