# Peer review of "Twice-Split Phosphorus Application Alleviates Low Temperature Stress by Improving Root Physiology and Phosphorus Accumulation, Translocation, and Partitioning in Wheat"

_agronomy, doi:10.3390/agronomy13102643_

Round 1
Reviewer 1 Report
Introduction
This section is informative and coincided with the manuscript subject. However, it can be improved by the following:
Use the following citation to support the statement "Numerous studies have shown that reactive oxygen species (ROS) accumulation and cell membrane disruption are the leading causes of wheat plant damage [14, 15]":
https://doi.org/10.15835/nbha51313291
Use the following citation to support the statement "Phosphorus application increases water and nutrient uptake by the root system [21]:
https://doi.org/10.1007/s10343-023-00843-2
https://doi.org/10.1080/03650340.2019.1619078
The author mentioned that " Based on the results of the previous studies, optimizing phosphorus application can be an effective………………". What are these studies?
Mention the hypothesis of the research at the end, and before aims, of introduction section
Materials and Methods
What is the number of replication used in the experimental design?
How many hours or days used for subjecting to low temperature stress?
What is the type of phosphorus fertilizer used and the active ingredient percentage?
What are the other factors (relative humidity, light, …….) inside artificial climate chamber used?
How long did the wheat plant spend in the climate chamber?
What is the time between ending the climate chamber and measurements?
Discussion
Under the subtitle:
4.2 Effects of Optimizing Phosphorus Application on Yield and Its Components
Use the following citation to support the statement " Phosphorus uptake and crop utilization are vital in determining the final crop yield[36]:
https://doi.org/10.1080/00103624.2019.1635147
Conclusions
Give an advice to the growers who grow wheat in cold areas based on the manuscript findings
Author Response
Introduction
This section is informative and coincided with the manuscript subject. However, it can be improved by the following:
Use the following citation to support the statement "Numerous studies have shown that reactive oxygen species (ROS) accumulation and cell membrane disruption are the leading causes of wheat plant damage [14, 15]":
https://doi.org/10.15835/nbha51313291
Answer: Thank you for your suggestion. We have used your suggested citation into the latest manuscript and made some modifications.
Use the following citation to support the statement "Phosphorus application increases water and nutrient uptake by the root system [21]:
Answer: Thank you for your suggestion. We have used your suggested citation into the latest manuscript and made some modifications.
The author mentioned that " Based on the results of the previous studies, optimizing phosphorus application can be an effective………………". What are these studies?
Answer: Thank you for your kind remind. We have supplemented these references in the latest manuscript:
https://doi.org/10.3389/fpls.2022.807844
https://doi.org/10.3390/agronomy12071700
https://doi.org/10.1016/j.jia.2023.09.013
Mention the hypothesis of the research at the end, and before aims, of introduction section
Answer: Thank you for your kind remind. We have supplemented the hypothesis in the latest manuscript:
In the present study, we hypothesized that the twice-split phosphorus application would enhance the wheat resilience to LT stress during the anther interval stage, improve root physiology, and increase phosphorus accumulation and final yield.
Materials and Methods
What is the number of replication used in the experimental design?
Answer: Thank you for your kind remind. In this experiment, a total of 72 wheat pots were planted, which means 18 pots per treatment. And all measurement indicators are repeated three times. We have added this to our latest manuscript.
How many hours or days used for subjecting to low temperature stress?
Answer: Thank you for your kind remind. The low temperature stress in this experiment was on the day of the anther interval stage and lasted 4 hours. We have added this to our latest manuscript.
What is the type of phosphorus fertilizer used and the active ingredient percentage?
Answer: Thank you for your kind remind. Phosphorus fertilizer type is superphosphate with not less than 12% active ingredient. We have added this to our latest manuscript.
What are the other factors (relative humidity, light, …….) inside artificial climate chamber used?
Answer: Thank you for your kind remind. The LT treatment was set at -4°C in the artificial climate chamber with a humidity of 75% and light intensity of 0 µmol·m−2·s−1·s from 1:00 a.m. to 5:00 a.m. We have added this to our latest manuscript.
How long did the wheat plant spend in the climate chamber?
Answer: Thank you for your kind remind. Wheat pots were stored in the climatic chamber for 4 hours for LT treatment.
What is the time between ending the climate chamber and measurements?
Answer: Thank you for your question. At the end of the LT treatment, we moved the potted plants out of the climatic chamber while taking samples (wheat roots) immediately. The wheat root samples were frozen in liquid nitrogen and then sealed in an ultra-low temperature refrigerator with tin foil to protect them from light. We had measured root physiological indicators in three days.
Discussion
Under the subtitle:
4.2 Effects of Optimizing Phosphorus Application on Yield and Its Components
Use the following citation to support the statement " Phosphorus uptake and crop utilization are vital in determining the final crop yield[36]:
https://doi.org/10.1080/00103624.2019.1635147
Answer: Thank you for your suggestion. We have used your suggested citation into the latest manuscript and made some modifications.
Conclusions
Give an advice to the growers who grow wheat in cold areas based on the manuscript findings
Answer: Thank you for your kind remind. We have added this section to the latest manuscript based on the content of the article and production practices:
This research will offer wheat farmers practical insights to mitigate abiotic stress and improve irrigation and fertilizer management at critical phases. In future research, it is necessary to explore measures to improve the combination of fertilizer effectiveness and wheat stress tolerance.
Thank you again for your valuable suggestions on this article.
Reviewer 2 Report
Dear Editors and Authors,
I read with interest the manuscript entitled “Twice-split Application of Phosphorus Alleviates Low Temperature Stress by Improving Root Physiology and Phosphorus Accumulation, Translocation, and Partitioning in Wheat”, in which sought to investigate the effects of the twice-split application of phosphorus on growth and development under LT stress during the anther interval stage, focusing on the antioxidant properties of the root system, dry matter and phosphorus accumulation, translocation and distribution of the wheat plants.. The subject of the article is important and has great relevance for the scientific environment of the study area. Therefore, the manuscript needs some adjustments so that it can then be forwarded to the publication process. The manuscript has the potential for publication in this journal Agronomy and needs the following adjustments:
ABSTRACT
- Add information about the analyzes carried out. In the section on material and methods.
- Replace keywords that are repeated in the title. Add short words. There are broad terms that need to be narrowed down.
INTRODUCTION
- There are paragraphs of great length. I suggest reducing it to better understand the information for readers of the manuscript.
- Reorganize this objective. It's very broad. There is no need to mention the analyzes that will be carried out. Add hypotheses for work at the beginning of this paragraph.
MATERIAL AND METHODS
- Add equations to explain the parameters related to phosphorus accumulation and remobilization.
- It is not necessary to mention the program used to make the Figures.
RESULTS
- Wouldn't it be possible to do this analysis on the figures in a factorial scheme, with lowercase and uppercase letters? If possible, I suggest you do it. Also add this change to the Material and Methods section.
- Replace Tables 1 and 2 with Figure. Visualization and standardization of results is improved.
DISCUSSION
- This section needs to be supplemented. There is important information, however, it is necessary for the authors to add more information discussing this work.
CONCLUSION
- Quote the LT temperature.
Author Response
ABSTRACT
- Add information about the analyzes carried out. In the section on material and methods.
Thanks for your constructive review. Information about carried analysis has been added to the Abstract section. (Highlighted)
Analysis of root physiology (enzymatic activities and acid phosphatase, contents of malondialdehyde, soluble sugar and soluble protein), phosphorus and dry matter accumulation, translocation, partitioning and agronomic and yield-related components was carried during this research study.
- Replace keywords that are repeated in the title. Add short words. There are broad terms that need to be narrowed down.
Answer: Thank you for your kind remind. We have made changes based on your suggestions in our latest manuscript.
Keywords: wheat; low temperature; optimizing phosphorus application; root physiology; phosphorus accumulation
INTRODUCTION
- There are paragraphs of great length. I suggest reducing it to better understand the information for readers of the manuscript.
Answer: Thanks for your constructive suggestions, lengthy paragraphs are shortens accordingly. (Highlighted)
- Reorganize this objective. It's very broad. There is no need to mention the analyzes that will be carried out. Add hypotheses for work at the beginning of this paragraph.
Answer: Thank you for your suggestion. We have reorganized the objectives in the latest manuscript while adding working hypotheses:
In the present study, we hypothesized that the twice-split phosphorus application would enhance the wheat resilience to LT stress during the anther interval stage, im-prove root physiology, and increase phosphorus accumulation and final yield. The ob-jectives of this study were to (1) examine the effects of twice-split phosphorus applica-tion on wheat roots' physiological indicators, such as antioxidant capacities under LT stress during the anther interval stage; (2) focus on the effects of various treatments on the accumulation, translocation, and distribution of dry matter and phosphorus in the wheat plants; and (3) explain the beneficial effects of twice-split phosphorus application on wheat yield and its components.
MATERIAL AND METHODS
- Add equations to explain the parameters related to phosphorus accumulation and remobilization.
Answer: Thanks for your constructive review. In this manuscript, due to the extensive treatment and parameters involved, there are already many abbreviations. We also made equations “PA= DMW × PC/ PT = (PA)f – (PA)m…”. But it doesn't seem to have much effect on improving the readability of the article. If equations are added to the latest manuscript, more abbreviations will be added, which is not conducive to readers' understanding. Therefore, we hope that the reviewers can understand our considerations. Thank you again.
- It is not necessary to mention the program used to make the Figures.
Answer: Thank you for your kind remind. In the latest manuscript, we have removed the content related to the diagramming procedure.
RESULTS
- Wouldn't it be possible to do this analysis on the figures in a factorial scheme, with lowercase and uppercase letters? If possible, I suggest you do it. Also add this change to the Material and Methods section.
Answer: Thank you for your kind remind. The current method of analysis is effective in presenting data and facilitating reader understanding. In keeping with most writing and analytical conventions, we use lowercase letters for analysis and description. We would like to discuss this issue with you in more depth. Thank you, again.
- Replace Tables 1 and 2 with Figure. Visualization and standardization of results is improved.
Answer: Thank you for your kind remind. This manuscript has 6 figures and 2 tables. Tables enable readers to see the size of each data at a glance. Due to the complexity of calculating the accumulation and remobilization of dry matter and phosphorus, tables can be used for clearer analysis and reading. In the latest manuscript, the figures and tables are more visualized side by side.
DISCUSSION
- This section needs to be supplemented. There is important information, however, it is necessary for the authors to add more information discussing this work.
Answer: Thank you for your kind remind. We have added more information to the latest manuscript for discussion.
4.1
This may be due to the improvement of soil microbial activity through optimizing phosphorus application [4]. Similarly, previous studies have indicated that delaying the application time of slow-release fertilizer increased the soil's enzyme activity, fertility and root activity [35].
LT stress disrupts the antioxidant system of the crop, resulting in MDA accumulation and toxicity [36]. At the same time, crops have evolved a range of physiology and metabolism to mitigate this damage in their natural environment [37, 38].
4.2
It has been shown that the application of additional urea before LT stress at the seedling stage of wheat was conducive to promoting the uptake of fertilizers by the root system, increasing the aboveground dry matter weight, and restoring wheat growth [42].
Wang et al. [43] pointed out that phosphorus storage and redistributive utilization among different organs can also effectively plant phosphorus homeostasis. And senescent leaves can serve as a major source of nutrients such as phosphorus during the later stages of wheat plant nutrient and reproductive growth [44-46].
4.3
Phosphorus uptake and crop utilization are vital in determining the final crop yield [47-49]. And optimizing phosphorus application can be a suitable agronomic and economical means of improving crop yield and nutrient accumulation, contributing to food and nutritional security [50, 51]. he main reasons for yield loss include decreased leaf photosynthetic capacity, reduced dry matter accumulation, and spikelet degradation [57-59].
This is because phosphorus fertilizer, an important nutrient for wheat growth and development, can also reduce spikelet degradation and increase spikelet setting by improving the antioxidant properties of the young spikes [4].
CONCLUSION
- Quote the LT temperature.
Answer: Thank you for your kind remind. Please refer to the latest manuscript
Thank you for your valuable suggestions on this article.
Reviewer 3 Report
Please explain the reasons for choosing the wheat variety as the subject of this experiment. Do you have any previous information to support this choice (different resistance to abiotic stresses)?
It is advisable to provide visualization of the object.
It is advisable to perform ROS analysis at low temperature voltage before and after phosphorus addition.
In "Conclusions" section please describe the prospects. Please indicate the importance and potential use of the obtained results.
The list of the references must be formatted according to the requirements of the journal.
Author Response
Please explain the reasons for choosing the wheat variety as the subject of this experiment. Do you have any previous information to support this choice (different resistance to abiotic stresses)?
Answer: In this experiment, Yannong19 was selected as the test subject. Yannong19 is the main cultivar in the wheat area of the Huang-Huai-Hai Wheat Region, and its annual area is among the top ten in China. In our previous study, low-temperature stress, especially freezing below 0°C, caused large yield (about 50%) losses of Yannong19. Meanwhile, previous studies have focused on photosynthetic limitation and spikelet degradation. In this study, the link between root physiology, material transport and yield were investigated in terms of optimizing phosphorus fertilizer application. We hope to start with the most basic fertilizer application to reduce the yield loss caused by low temperature stress.
https://doi.org/10.3390/plants11030389
https://doi.org/10.3389/fpls.2022.811884
It is advisable to provide visualization of the object.
Answer: Thank you for your kind remind. We have reorganized the objectives in the latest manuscript while adding working hypotheses:
In the present study, we hypothesized that the twice-split phosphorus application would enhance the wheat resilience to LT stress during the anther interval stage, im-prove root physiology, and increase phosphorus accumulation and final yield. The objectives of this study were to (1) examine the effects of twice-split phosphorus application on wheat roots' physiological indicators, such as antioxidant capacities under LT stress during the anther interval stage; (2) focus on the effects of various treatments on the accumulation, translocation, and distribution of dry matter and phosphorus in the wheat plants; and (3) explain the beneficial effects of twice-split phosphorus application on wheat yield and its components.
It is advisable to perform ROS analysis at low temperature voltage before and after phosphorus addition.
Answer: Thank you for your kind remind. We apologize for not sampling the wheat roots before the low temperature voltage. Your suggestion makes a lot of sense. In future studies, we will perform ROS analysis before low temperature voltage. Thanks again.
In "Conclusions" section please describe the prospects. Please indicate the importance and potential use of the obtained results.
Answer: Thank you for your kind remind. We have added this content to our latest manuscript.
This research will offer wheat farmers practical insights to mitigate abiotic stress and improve irrigation and fertilizer management at critical phases. In future research, it is necessary to explore measures to improve the combination of fertilizer effectiveness and wheat stress tolerance.
The list of the references must be formatted according to the requirements of the journal.
Answer: Thank you for your kind remind. We have made changes to comply with the journal's requirements regarding references. Thank you again.
Thank you again for your valuable suggestions on this article.
Round 2
Reviewer 2 Report
Dear,
The authors made the suggestions previously proposed to improve the work.